# Sustainable Synthesis of Zeolitic Imidazolate Frameworks at Room Temperature in Water with Exact Zn/Linker Stoichiometry

**DOI:** 10.3390/nano14040348

**Published:** 2024-02-12

**Authors:** María Asunción Molina, Jorge Rodríguez-Campa, Rosa Flores-Borrell, Rosa M. Blanco, Manuel Sánchez-Sánchez

**Affiliations:** 1Instituto de Catálisis y Petroleoquímica (ICP), CSIC, C/Marie Curie 2, 28049 Madrid, Spain; asuncion.molina@ucl.ac.uk (M.A.M.); j.rodriguezcampa@alumnos.urjc.es (J.R.-C.); mr.flores@alumnos.urjc.es (R.F.-B.); rmblanco@icp.csic.es (R.M.B.); 2Department of Chemistry, University College London, 20 Gordon Street, London WC1H 0AJ, UK; 3Research Complex at Harwell, Rutherford Appleton Laboratory, Didcot OX11 0A, UK

**Keywords:** sustainable synthesis, deprotonating agents, ZIF-8, ZIF-67, exact linker/Zn ratio, amines, g-C_3_N_4_-like species

## Abstract

Zeolitic imidazolate frameworks (ZIFs) are widely used MOFs because of certain characteristics, but also because they can be prepared at room temperature using water as the unique solvent. However, these a priori sustainable conditions inevitably entail a huge and somehow unusable excess of linker. Here, we present the formation of ZIFs at room temperature in water, starting from mixtures with a linker/metal ratio of two, that is, coinciding with the stoichiometry found in the final MOFs, in the presence of amines. ZIF-8 can be prepared with triethylamine (TEA), giving a yield of Zn of 96.6%. Other bases, like NaOH, tetraethylammonium hydroxide or ammonium hydroxide, do not lead to ZIF-8 under the same conditions. The so-obtained ZIF-8 contains TEA inside its cavities, making it less porous than its conventionally prepared counterparts. Amine can be removed by mild thermal treatments (200–250 °C). Such thermal treatments induce the generation of g-C_3_N_4_-like species which could give added value to these materials as potential photocatalysts, increasing their affinity to CO_2_, as proved in this work. This methodology can be successfully extended to other amines, like N,N-dicyclohexylmethylamine, as well as to other prepared ZIFs, like Co-based ZIF-67, isostructural to ZIF-8.

## 1. Introduction

Amongst the almost countless number of MOFs [1], the zeolitic imidazolate frameworks (ZIFs) family [2] has a few significant peculiarities. Firstly, in clear contrast to carboxylate-based MOF materials, the metal ions (Ms) of ZIFs do not form clusters as they are isolated; as a consequence, they do not possess M-M, M-O-M or M-N-M bonds, but instead, they possess singular [M^II^(linker)_4_]^2−^ environments. Secondly, their structures are zeolitic but much more open than their structurally homologous zeolites, as the single oxygen atoms linking contiguous silicon atoms in zeolites are substituted by organic ligands linking contiguous metals in ZIFs. Thirdly, related to the two previous differences, their metals generally have tetrahedral coordination as a consequence of the crystallographic imposition of zeolitic structures. This means, on the one hand, that the metals cannot be, strictly speaking, open metal sites, but their 4-fold coordination is potentially and reversibly increased up to either 5- or 6-fold in the presence of an adequate adsorbate/reactant with no structural transformation, for instance, Ti or other heteroatoms in zeolites [3]. On the other hand, the variety of metals forming ZIFs is rather limited as not many metals can comfortably adopt tetrahedral coordination as a base for 3D structures, considering that the metals should, in principle, be divalent. This makes Zn and Co the most common metals forming ZIFs so far; although, some others such as Fe, Cu, Cd or Mn can also eventually form ZIF materials [4]. Finally, their linkers are reduced to imidazolates, as they are the only ones capable of simultaneously being bidentate, monovalent and with N atoms in an ideal disposition to give an angle of M-linker-M (145°), similar to that given by Si-O-Si in zeolites [2]. One of the immediate consequences of this fact is that metals are exclusively surrounded by (four) N atoms.

All these specifics make ZIFs special MOFs [5]. They are being intensively investigated in applications such as energy storage [6], electrocatalysis [7], heterogeneous catalysis [8], drug delivery [9], enzyme support [10], environmental remediation [11] or gas separation [12], among others. Nevertheless, another key factor of such widespread use of ZIFs is their synthesis under benign conditions, at room temperature and with water as the solvent. In this context, it is not unrelated that sod-structured Zn-based ZIF-8 [2] (and its Co-based counterpart ZIF-67 [13]), which is the most suitable ZIF material to be prepared under these conditions, is by far the most applied ZIF material, the quintessential ZIF, in spite of certain intrinsic structural restrictions (their cavities are accessible to windows with a diameter as small as 3.8 Å, although they are quite flexible [14]).

Although the above-mentioned benign synthesis conditions, that is, water as the unique solvent and at room temperature, seem to guarantee sustainable synthesis, the truth is that nowadays, it is far from being completely sustainable due to the need to start with a massive excess of linker (with a linker/Zn ratio of about 40 and never below 20) [15,16,17]. That ratio can be reduced to eight when water is substituted by other solvents (different alcohols or acetone but not, for instance, N,N-dimethylformamide) [18,19]. Even the synthesis of ZIF-8 at room temperature in water with a linker/Zn stoichiometry of two found in the crystallized material has been reported, but in the presence of a complicated mixture of surfactants [20]. Alternatively, ZIF-8 can be prepared under sustainable conditions in the presence of triethylamine still with an excess of linker [21,22,23,24] or with a huge amount of ammonium hydroxide (NH_4_OH/linker = 16) [25,26]. Other amines have been used in the synthesis of ZIF-8 starting with an excess of Zn [27]. Other quite sustainable methodologies for the preparation of ZIF-8 have been also published elsewhere [28,29,30], but they require mechanical energy input. All the above-mentioned applications of ZIF-8 could be intensified and even expanded if a real/completely sustainable method of synthesis of this material is discovered. The challenge continues.

One of the most successful strategies to synthesize carboxylate-based MOFs under completely sustainable conditions has been developed by our group, and consists of the use of salts as linker precursors or, the same as in water, using a stoichiometric amount of deprotonating agents [31,32,33,34,35,36,37]. This leads to a synthesis methodology of MOFs under conditions that are almost impossible to improve from a sustainability point of view, including no energy input, room temperature, water as the unique solvent, low cost of the reactants, high availability of the reactants, low toxicity, no generation of (toxic) by-products, quick kinetics, high yield, no special bakers, high-quality MOFs and a stoichiometry process, the latter being a very pertinent characteristic for the aim of this work. In addition, the resultant, immediately formed MOF materials are generally nanocrystalline, as nucleation predominates against crystal growth, which makes these sustainable MOFs very attractive for their use as direct heterogeneous catalysts [38,39], as solid support for immobilizing enzymes [35,40,41,42,43,44] or as the base of composites where the interaction between the individual components is prioritized [36,37]. Any tested base, such as NaOH, NH_4_OH or amines (in particular, triethylamine), irrespective of their basic strength, leads to the deprotonation of the linker and the subsequent formation of the corresponding MOF as soon as they are offered a metal in solution [40]. On the contrary, 2-methylimidazole (H-2-mIM), the most extended imidazole linker in ZIFs, is quite soluble in water, which allows for ZIF-8 [15] (and ZIF-67 [45] or ZIF-L [46], among others) to be prepared at room temperature in aqueous solutions. But soluble does not mean deprotonated. Its pKa is around eight [47], so its deprotonated form 2-mIM^-^ in distilled water is not dominant. This would justify the formation of ZIFs at room temperature and in water, but only under a large excess of linker in the synthesis media, in order to offer the metal a sufficient amount of deprotonated species. Starting from a higher pH should facilitate its kinetics and its yield as well as avoiding the use of an excess of linker.

In summary, the challenge faced in this work is to prepare ZIF-8 (in general, ZIFs) under truly sustainable conditions with a special emphasis on starting from a mixture having the linker/Zn ratio equal to that found in the final structure of ZIF-8.

## 2. Materials and Methods

### 2.1. Synthesis and Thermal Treatments of ZIF Materials

A conventional ZIF-8 was prepared following the protocol reported by Kida et al. [16], since ZIF-8 can be prepared from mixtures with the lowest linker/Zn ratio (20) reported at room temperature and in water. ZnCl_2_ was used as Zn precursor. The molar composition of the mixture was: 1 ZnCl_2_: 2 H-2mIm: 450 H_2_O and the resultant sample was called ‘No base’.

Taking this synthesis as a starting point for our approach, different at-room-temperature preparation attempts of ZIF-8 from aqueous solutions containing a 2mIM/Zn ratio of 2 were carried out in the presence of the molar amount of different bases (base/linker ratio of 1): tetraethylammonium hydroxide (TEAOH, 35 wt.% in aqueous solution), NaOH, triethylamine (TEA) and ammonium hydroxide (NH_4_OH, 25 wt.% in aqueous solution). The samples were named following the nomenclature ‘Base name_base/linker ratio’, that is, their names were TEAOH_1, NaOH_1, TEA_1 and NH_4_OH_1. In these syntheses, 1.23 g of ZnCl_2_ (9 mmol) was dissolved in 15 g of distilled water (pH = 6.0); meanwhile, 1.48 g of 2-methylimidazole (18 mmol) was dissolved in 52.35 g of distilled water in the presence of 18 mmol as base, either tetraethylammonium hydroxide (TEAOH, 35 wt.% in aqueous solution), NaOH, triethylamine (TEA) or ammonium hydroxide (NH_4_OH, 25 wt.% in aqueous solution). The pH levels of the clear solution of the linker in the presence of a base were 13.7, 12.7, 12.4 and 11.7. Obviously, the order of the pH is a consequence of the base strength, but in any of these cases, the linker (pKa ~ 8.0 [47]) would be mostly deprotonated. Once both mixtures became clear solutions (in a few minutes), the metal solution was added in drops to the solution of the linker, which provokes the immediate appearance of abundant white solids. The molar composition was: 1 ZnCl_2_: 2 H-2mIm: 2 Base: 450 H_2_O. The pH levels of these mixtures were 12.4 for the mixture prepared with TEAOH, 11.8 for the mixture prepared with NaOH, 10.1 for the mixture prepared with TEA and 9.5 for the mixture with NH_4_OH, which suggests that the linker should be in its deprotonated form even after being mixed with the metal solution. The mixture was kept stirring at room temperature during 20 h; then, the solid was recovered by centrifugation, washed three times with water and dried at 100 °C.

The rest of the syntheses presented in this work were prepared following the same procedure, but just changing either the base/linker ratio, the nature of the amine and/or the nature of the metal. Thus, a series of TEA_n samples (where n is the base/linker ratio) were prepared through modifying the content of TEA in the synthesis media; N,N-dicyclohexylmethylamine (MCHA) was used as a bulkier amine against TEA in an attempt to prepare ZIF-8 (sample MCHA_1.5) and Co was used instead of Zn for attempting the synthesis of other ZIFs, in particular ZIF-67, with both mentioned amines. These samples, which were purple in color, were called Co_TEA_2.2 and Co_MCHA_1.5 as a lower base/linker ratio than for TEA was optimized for MCHA.

The sample TEA_2.2 was heated at different temperatures (200, 225 and 250 °C) in a conventional oven for 20 h. The resultant samples were denoted as TEA_2.2_200C, TEA_2.2_225C and TEA_2.2_250C.

The yields of the synthesis of ZIF-8 were calculated from TGA. The residual weight after the TGA heating treatment under air flow was ZnO, as it was checked by XRD characterization. The corresponding Zn amount was divided by the added amount of Zn into the synthesis mixture and multiplied by 100 to obtain the % yield.

### 2.2. Characterization of ZIF Materials

Powder X-ray diffraction was carried out using a PANalytical X’pert Pro Instrument (Cu Kα radiation). Thermogravimetric analyses (TGA) were performed in a Perkin-Elmer TGA7 instrument, in air flow (40 mL/min) and 20 °C/min of heating rate from room temperature to 900 °C. Thermogravimetric analysis–mass spectrometry (TGA-MS) was carried out on a TA Instruments Q500 thermobalance coupled to a mass spectrometer Omnistar GSD 301 O/301T of ThermoStar. Field-emission–scanning electron microscopy (FE-SEM) was carried out using high-resolution Hitachi SU8000 equipment. Nitrogen adsorption isotherms were measured in a Micromeritics ASAP 2420 apparatus at the temperature of liquid N_2_ (−196 °C). The samples were degassed in situ at 150 °C in vacuum for 16 h prior analysis. The surface area was determined using the BET method. CO_2_ isotherms were registered in Micromeritics ASAP 2020 equipment at 0 °C, in the partial pressure interval of 0.0–0.03, because the vapor pressure of CO_2_ at that temperature is 0.03. UV–visible diffuse reflectance spectroscopy (UV-vis DRS) measurements were run in a Varian Cary 5000 double-beam UV-vis Near-IR spectrophotometer. The ^1^H-to-^13^C (100.61 MHz) CP (cross-polarization) MAS (magic-angle spinning) nuclear magnetic resonance (NMR) spectra were registered on a Bruker AV WB 400 spectrometer at 25 °C using a 4 mm triple channel probe head, spinning the sample at 10 kHz and with an acquisition time of 0.05 s. The excitation ^1^H pulse was of 3 µs, contact time of 3 ms, time between consecutive pulses of 4 s and decoupling tppm15-type at 80 kHz. Chemical shift δ^13^C was referenced with respect to tetramethylsilane TMS (0 ppm).

## 3. Results

### 3.1. Sustainable Synthesis of ZIF-8

This work involves testing different deprotonating agents at temperature and synthesis of the ZIFs starting from aqueous solutions that have linker/Zn ratios corresponding to the stoichiometry of the final ZIF, with particular emphasis on ZIF-8. Figure 1A shows the XRD patterns of the solids obtained from different attempts at ZIF-8 preparation in the presence of four bases (base/linker ratio of 1): in order of basicity strength, TEAOH ~ NaOH > TEA > NH_4_OH. According to the pHs given in the Section 2, all bases are strong enough to lead to a final mixture with a pH (range 12.4–9.5) at which the linker is mainly in its imidazolate form, which is required for forming ZIF-8. Conventional ZIF-8 shown in Figure 1 was prepared starting from a mixture with a linker/Zn ratio of 20 [16], ten times higher than the stoichiometry in ZIF-8. If the linker/Zn ratio of the mixture is 2, coinciding with that of ZIF-8, then the Unknown phase 1, which has been reported elsewhere [19], is formed. In spite of the diffractogram, this phase contains a relatively low angle peak (2Θ ~ 9.4°), and its porosity is quite reduced (16 m^2^g^−1^; Figure A1). The presence of any of the test bases altered the nature of the formed phase, which indicates a relevant role for the basicity of the mixture in the phase selectivity. However, none of the tested bases were able to provide the formation of the pure ZIF-8 material. Indeed, none of the diffractograms prepared with TEAOH, NaOH or NH_4_OH showed any indication of the ZIF-8 phase. The phases found in the samples NaOH_1 and NH_4_OH_1 have been already detected elsewhere in ZIF-8 preparation attempts at room temperature in water when the starting mixture is too dilute [17], so there may also be a chance to prepare ZIF-8 with these bases. At this point, it must be noted that this ‘deprotonating’ strategy was successful in the preparation of carboxylate-based MOFs irrespective of the nature of the base, including NaOH, NH_4_OH or TEA [40]. Only the diffractogram of the sample TEA_1 has peaks that correspond to ZIF-8 (the reflections 110 and 200 are marked to highlight the unequivocal presence of that phase). The reason why TEA, and not the rest of the test bases, are able to facilitate the formation of ZIF-8 is not clear. Obviously, it is not a matter of basicity strength and probably not a matter of structure directing effect, since ZIF-8 can be formed in the absence of TEA (although an excess of linker is necessary). In any case, the presence of the Unknown phase 1 is also evident in the sample TEA_1. Since Unknown phase 1 is the only phase detected in the absence of any base (sample ‘No base’), it is presumed that a higher amount of TEA could lead to pure ZIF-8.

Figure 1B compiles the XRD patterns of the different TEA_n samples, where n denotes the TEA/linker ratio. As expected, aqueous mixtures with TEA content below a TEA/linker ratio of 2 scarcely produced ZIF-8. To detect traces of this MOF, at least a TEA/linker ratio of 0.75 was necessary. Following this trend, increasing TEA content in the starting mixture makes the formed solid richer in ZIF-8. A TEA/linker ratio of 2.2 was required to achieve completely impurity-free ZIF-8 samples. That is why the sample TEA_2.2 was selected for further characterization. It is important to highlight that the yield in Zn of this experiment was 96.6%, so practically all Zn present in the initial mixture (and presumably also all linker, as the linker/Zn ratio of the starting mixture was 2 and no impurity was detected) ended up in the final ZIF-8. Moreover, the formation of the solid is instantaneous and massive as soon as the sources of linker and metal are put together, which reinforces the linker deprotonation role of the amine. It contrasts with the cloudy formation of the conventional ZIF-8 [16]. Figure A2 shows FE-SEM images of the conventional ZIF-8 and the sample TEA_2.2. The shape of the crystals of the conventional ZIF-8 follows a pattern, with the typical rhombic dodecahedron shape of this methodology (Figure A2A) [16]. However, it is hard to find any pattern in the shape of the crystals forming the sample TEA_2.2 (Figure A2B), probably because its formation is instantaneous, and then nucleation presumably dominates over crystal growth, leading to shapeless crystals.

One particular feature of all XRD patterns in the samples containing ZIF-8 in Figure 1B is the relatively low intensity of the XRD reflection 110, the first appearing at the lowest 2Θ angle, which does not become the most intense, as usually happens (see for instance the XRD pattern of the conventional ZIF-8 in the same figure). Since the sod topology is cubic, the preferred orientation of the crystals is not probable, and therefore some other reason must be behind this behavior. Given that the XRD reflection 110 is related to the diffraction phenomenon along the cavities, it is probable that its intensity decrease is due to some adsorbed molecules.

To shed light on this XRD feature, some samples were characterized using thermogravimetric analysis (TGA) (Figure 2). The TGA curve of the conventional ZIF-8 shows two well defined weight losses at about 200 °C, which are attributed to some protonated unreacted linker (H-2mIM) in good agreement with the TGA of the commercial linker (gray line) and with the literature [16]; the main loss which started around 400 °C, which is attributed to the organic linker bound to Zn^2+^ ions in ZIF-8. The TGA of the sample TEA_2.2 has also two weight losses. The one at higher temperatures is obviously also due to the linker decomposition of ZIF-8. However, the weight loss at lower temperatures cannot be assigned to the protonated linker, as it is shifted 100 °C with respect to the loss in TGA of conventional ZIF-8. It must be noted that both the absence of any excess of linker in the starting mixture of our protocol and the more basic pH at which the synthesis is carried out would prevent the presence of any protonated linker in the final ZIF-8 sample, which is a significant impurity in the ZIF-8 samples prepared by this conventional method, in particular when water is reduced to minimize the huge excess of linker [16].

To clarify the assignment of the weight loss around 280 °C, a TGA-mass spectrometry (Figure A3) and a solid-state ^1^H-to-^13^C CP MAS NMR (Figure 3) of the sample TEA_2.2 were run. In the temperature region of this weight loss (Figure A3), the mass spectrometry shows three particular m/e signals, which are not present in the conventional ZIF-8 sample. These m/e were 72 and 86 and 101, which coincide with the mass of the species N(CH_2_CH_3_)_2_, NCH_2_(CH_2_CH_3_)_3_ and N(CH_2_CH_3_)_3_, the latter matching with triethylamine whilst the other two are perfectly compatible with fragments coming from the TEA decomposition under the ion current applied during the acquisition. In addition, the three m/e signals have the same patterns as a function of temperature, indicating their common origin. The ^1^H-to-^13^C CP MAS NMR spectra (Figure 3) indicated the presence of the two ^13^C resonances of triethylamine at δ^13^C of 11.58 and 46.45 ppm (tabulated δ^13^C for TEA in CDCl_3_ are 11.78 and 46.46 ppm) in the sample TEA_2.2, which are absent in the conventional ZIF-8. Moreover, the low intensity and high narrowness of these two signals in comparison with those of the linker in the spectra indicate that TEA possesses very high mobility as the ^13^C NMR spectra have been registered under a cross polarization pulse [48]. Nevertheless, the TEA molecules must be inside the sod cavities of ZIF-8; otherwise, the fact that they were not removed during the washing protocol cannot be explained especially as their weight loss (centered at ca. 280 °C; Figure 2) has taken place at a temperature almost 200 °C higher than its boiling point (89.3 °C). By the way, the TEA molecules inside the ZIF-8 cavities also justify the relatively low intensity of the XRD reflection 110 in ZIF-8 samples prepared with TEA (Figure 1).

In summary, TGA-MS and ^1^H-to-^13^C CP MAS NMR characterization confirm that the ZIF-8 material formed under sustainable conditions in the presence of TEA contains a significant amount of this amine within its pores, which cannot be released unless it is previously decomposed. It is logical since its kinetic diameter (7.8 Å [49]) is much larger that the entrances windows (3.8 Å) to the ZIF-8 cavities.

Far from considering the entrapped amine within the ZIF-8 cavities strictly as a drawback, two key questions should be asked: (i) if the amine can be removed without altering the structure and the porosity of the materials, our method becoming a new sustainable route to achieve ZIF-8; and, not least, (ii) if this easily obtained TEA-containing ZIF-8 could find singular applications unattainable for conventional ZIF-8. It must be noted that entrapping this amine within ZIF-8 by any post-synthesis treatment is not possible due to steric hindrance, as the entrapping during the formation of the MOF is instantaneous and in one step.

To answer the first question, Figure 4 compares the N_2_ adsorption/desorption isotherms of the conventional ZIF-8 with these of the TEA_2.2, both as the as-synthesized one and after being thermally treated at 200, 225 and 250 °C in a conventional oven with no atmosphere control (that is, in air). The BET surface area given by the conventional ZIF-8 (1577 m^2^g^−1^) is even slightly higher than that prepared under the same conditions and reported elsewhere (1480 m^2^g^−1^) [16]. The textural properties of the sample ZIF-8, prepared in the presence of TEA sample TEA_2.2, are rather limited (S_BET_ of 581 m^2^g^−1^), undoubtedly due to the entrapped TEA molecules. Indeed, the thermal treatment at 200 °C during 20 h removes all the TEA molecules from the sample TEA_2.2 (see Figure 2), and leads to a substantial increase in its textural properties (S_BET_ of 1380 m^2^g^−1^), not so far from those of the conventional ZIF-8 sample. The increase in temperature of the thermal treatment leads to a systematic reduction in the textural properties of the ZIF-8 material, reaching 1242 m^2^g^−1^ at 225 °C and 953 m^2^g^−1^ at 250 °C. Such a drop in the textural properties must be related to the stability of ZIF-8, which could start to suffer at these temperatures [50]. In this sense, the samples change from white to yellow, that is to say, they are darker the higher the heat treatment temperature.

The change in color tonality of the samples perceptible to the naked eye led us to study these samples by means of ultraviolet–visible diffuse reflectance spectroscopy (UV-vis DRS). The results are shown in Figure 5A. The UV-vis DRS spectra of the conventional ZIF-8 and TEA_2.2 samples, both white, almost overlap; the scarce difference between them could be attributed to the presence of the amine in the latter. The thermal treatment of the ZIF-8 leads to significant changes in the spectra, as expected from the color change to yellow. At least three new bands are formed, one of them clearly invading the visible region. The intensity of these bands increases with increasing the thermal treatment temperature; however, the intensity of the bands does not increase from the spectra of the sample TEA_2.2_225C to TEA_2.2_250C. These bands are quite similar in positions to those given by the well-known graphitic carbon nitride g-C_3_N_4_, which is also a yellow polymeric material consisting of C, N and some impurity H, connected via tris-triazine-based patterns, with extensive possibilities as metal-free photocatalysts [51]. For instance, it complements the potential photocatalytic of MOFs when both are part of a given composite [36,37]. The similarity in the UV-vis features of the thermal-treated ZIF-8 with those of g-C_3_N_4_ suggests that the still-porous thermally treated yellow ZIF-8 could have unexplored applications in (photo)catalysis.

To shed further light on the nature of the generated species in thermal-treated ZIF-8, Figure 5B presents the 145–175 ppm region of the ^1^H-to-^13^C CP MAS NMR spectrum of the sample TEA_2.2_225C (selected amongst the series of TEA_2.2_T because of its acceptable BET surface area, Figure 4, and the intensity of its generated UV-vis bands by thermal treatment, Figure 5) compared with that of the same samples before calcination, TEA_2.2, and that of the conventional ZIF-8. The most intense signal with δ^13^C at ca. 151 ppm, present in the three NMR spectra, is the lowest-field ^13^C signal of the linker 2-methyl-imidazolate of the ZIF-8 material (see Figure 3). Apart from that, the ^13^C NMR spectrum of the sample TEA_2.2_225C contains two new signals at δ^13^C 154.8 and 166.9 ppm, which are absent in the spectra of both the conventional ZIF-8 and the as-synthesized TEA_2.2 sample. Interestingly, the chemical shift δ^13^C of the g-C_3_N_4_ is quite close to that (157 and 165 ppm) [52], reinforcing the theory that the species forming during the thermal treatment of ZIF-8 are indeed of a very similar nature to the widely used graphitic carbon nitride g-C_3_N_4_. It must be noted that the huge difference in intensity between the ^13^C NMR peaks of the linker and these of the g-C_3_N_4_-like species is accentuated due to the type of NMR pulse sequence used. The cross polarization phenomenon is magnified in static (not-mobile) species and, more relevant in this case, in species having ^1^H isotopes near their ^13^C ones. g-C_3_N_4_ scarcely possesses ^1^H as impurities [51], whereas the ZIF-8 linker possess H within their own structure.

Beyond the potential photocatalytic behavior of the thermal-treated ZIF-8, the second big query concerning the TEA-containing ZIF-8 reported here is to find singular applications. In this context, it is well-known that amines have a particularly relevant affinity for CO_2_ [53], with the capture/adsorption/separation of this gas being probably one of the greatest challenges in the field of gas adsorption. On the other hand, ZIF-8 is one of the MOF materials most extensively tested in the adsorption and separation of CO_2_ [54]. Figure 6 displays the CO_2_ isotherms registered at 0 °C of the conventional ZIF-8, the sample TEA_2.2 and the sample TEA_2.2_225C. The CO_2_ isotherms have been registered in a partial pressure (p/p_0_) interval of 0–0.03 (0.03 is the partial pressure of CO_2_ at 0 °C, which is equivalent to 1.052 bar) as the volumetric equipment does not allow the atmospheric pressure to be overcome. That is why any of the tested ZIF-8 materials are far from saturated. Our conventional ZIF-8 is able to adsorb 58.1 mg CO_2_ per g under these conditions (0 °C), which exceeds some comparable data published elsewhere [55,56,57], although the highest porous ZIF-8s were able to uptake higher amounts of CO_2_ [58], indicating the key importance of the surface area of ZIF-8 for its CO_2_ uptake. In any case, the main interest of these materials is in studying their affinity for CO_2_, which can be studied at low pressure, that is, at low CO_2_ loading. It is significant that at very low pressure (below p/p_0_ = 0.005), the least-porous amine-containing ZIF-8 (sample TEA_2.2) is able to adsorb more CO_2_ than the other two more porous ZIF-8 samples, making it clear that the amine TEA has a positive effect on the adsorption of CO_2_. However, this effect only becomes relevant at low pressures; at higher pressures, the better textural properties of the other two ZIF-8 samples prevail. It must be noted that part of the ZIF-8 cavities are not accessible to CO_2_ in the sample TEA_2.2, not only because of the pore volume occupied by TEA but also because TEA molecules could block other void cavities. It is even more relevant that the thermally treated ZIF-8 far exceeds the CO_2_ adsorption capacity of the conventional ZIF-8 along all studied pressures, in spite of the fact that the BET surface area of the former is around 20% lower than that of the latter (Figure 4). Such CO_2_ uptake is much higher (around 50%) than that given by a very high porous ZIF-8 [58], whose BET surface area is 37% higher than our sample TEA_2.2_225C. This highlights that the species formed during the thermal treatment, presumably of carbon nitride nature, has indeed a very favorable effect on CO_2_ adsorption. It must be indicated that there is not any evidence that the species formed come from the decomposition of the amine but rather from the partial decomposition of the ZIF-8 linker. In fact, their ^13^C chemical shifts are characteristic of C atoms linked to N atom(s) within aromatic rings, whose origin more probably comes from the linker than from the amine. Moreover, the conventional ZIF-8 sample thermally treated under the same conditions also turns yellow and gives a similar UV-vis DRS spectrum to these of the sample TEA_2.2_T. It must be remarked that the initial decomposition of ZIF-8 by mild thermal treatment has been studied by other authors, always leading to the increased capacity of CO_2_ adsorption/uptake. Thus, Gadipelli et al. reported the breaking of the Zn-N bonds and the consequent formation of exposed metal sites when ZIF-8 is thermally treated above 400 °C under inert atmosphere (Ar); this led to a significant increase in CO_2_ uptake [59]. Furthermore, thermal treatments of ZIF-8 under similar conditions to these carried out in our work led to the appearance of species containing C=N-OH groups due to the partial oxidation of the linker, in good agreement with our interpretation, although the CO_2_ adsorption capacity of the resultant ZIF-8 was not improved [60].

### 3.2. Versatility of the Methodology: Preparation of ZIF-8 and ZIF-67 with Other Amines

So far, this new synthesis has only been presented for the sustainable formation of a ZIF material (ZIF-8) and with the use of one particular amine (TEA). In this section, the same methodology is used for the synthesis of the same MOF but in the presence of another amine (N,N-dicyclohexylmethylamine, MCHA) as well as for the preparation of other ZIF material, ZIF-67, which is isostructural of ZIF-8 but based on Co, with either TEA or MCHA as an amine. Figure 7 shows the XRD patterns of the ZIF-8 prepared with MCHA (sample MCHA_1.5) and the ZIF-67 prepared with TEA (sample Co_TEA_2.2) and with MCHA (Co_MCHA_1.5). The three XRD patterns show pure sod-structured ZIF. Different deductions can be extracted from these successful experiments. First of all, TEA is not a ‘magic’ amine in this protocol. Other amines, like MCHA, can have the same role in this method, unlike other bases, either stronger (NaOH or TEAOH) or weaker (NH_4_OH) (Figure 1). Given the difference in size and nature of the substituted group R of these two tertiary amines NR_3_, it seems that their role is not as structure-directing agents, contrasting to their role of the same amines in the synthesis of crystalline microporous aluminophosphate-based materials [61]. Moreover, the ideal linker/amine ratio (the minimum amount of amine for achieving pure ZIF) could be particular for any amine. Another important conclusion is that the methodology is extensible to other ZIFs, such as ZIF-67. The ZIF-67 so obtained is formed by slightly smaller crystals (Figure A2C) with the same shapeless morphology as those of the ZIF-8 TEA_2.2 (Figure A2B), again, probably as a consequence of having been formed by immediate precipitation.

On the other hand, the decrease in relative intensity of the lowest 2Θ angle peak (reflection 110, at 2Θ ~ 7.3°) with respect to the intensity of the rest of the peaks of the same XRD pattern of the samples ZIF-8/ZIF-67 prepared in the presence of the bulkier amine MCHA is greater than when TEA is the amine. It suggests that the MCHA molecules, like the TEA, are within the sod cavities. MCHA interferes more in the diffraction phenomenon for this particular reflection (whose diffraction signal necessarily has to pass through the sod cavities) than TEA because of the greater bulk of the former.

Figure 8A shows the TGA curves of the ZIF-8 prepared in the presence of MCHA and the two ZIF-67 prepared with TEA and MCHA. Just like the sample ZIF-8 prepared with the amine TEA (Figure 2), the XRD patterns (Figure 7) suggest these samples contain amines. Unlike the TGA of the sample TEA_2.2, the weight losses of the linker and the amine are overlapped as they occur at closer temperatures. In particular, the two ZIF-67 samples are much less stable than both the ZIF-8 samples, as is well-known from the literature [62]. This fact could complicate the removal of the amine by thermal treatment and maintaining intact the structure of the corresponding ZIF material.

The presence of the amine within the sod cavities obviously conditions the textural properties of these ZIF materials, as is made clear by the N_2_ adsorption isotherms shown in Figure 8B. The decrease in BET surface area is more accentuated in samples prepared with MCHA, again because of its bulkier nature and because they can presumably block the accessibility of N_2_ molecules to the cavities more efficiently than TEA. The much higher BET surface area of the sample Co_TEA_2.2 (1003 m^2^g^−1^, Figure 8B) compared to that of its Zn-based counterpart TEA_2.2 (581 m^2^g^−1^, Figure 4) is due to the much higher content of amine TEA in the latter (more than 20 wt.% of the total sample is amine; Figure 2) than in the former (less than 15 wt.%; Figure 8A, although this weight loss is overlapping that of the ZIF-67 linker 2-methylimidazole). The reason why ZIF-8 retains a higher amount of TEA than ZIF-67 is not yet clear.

## 4. Conclusions

A new and more sustainable method for preparing ZIF materials at room temperature and in water is presented by adding just amines to the synthesis media. The key to sustainability is the fact that it starts from a synthesis mixture having the same link-er/metal ratio as that found in the final ZIF, together with the higher yield. Other bases (NaOH, TEAOH or NH4OH) did not lead to the formation of ZIFs under the same conditions. In this way, ZIF-8 is immediately and easily formed, with a high yield (96.6% yield in Zn after 20 h) at room temperature and, more importantly, starting from an aqueous mixture with a linker/Zn ratio of 2 in the presence of triethylamine (TEA). The resultant ZIF-8 has a distinctiveness against the conventional ZIF-8 materials: it contains TEA molecules inside the sod cavities, as demonstrated by TGA-MS and ^1^H-to-^13^C CP MAS NMR. Since these TEA molecules cannot diffuse through to the windows giving access to the sod cavities, a possible way to free the pores is to thermally treat the sample at temperatures allowing the ZIF-8 to maintain its structure, with 200–250 °C being a good compromise. Accordingly, the thermally treated samples turn yellow because they contain some g-C_3_N_4_-like species, which makes them potential photocatalysts and increase their CO_2_ affinity. This methodology is extensible to the use of other amines, like the bulkier N,N-dicyclohexylmethylamine, and to prepare other ZIFs, even if they are based on Co, like ZIF-67.

## Figures and Tables

**Figure 1 nanomaterials-14-00348-f001:**
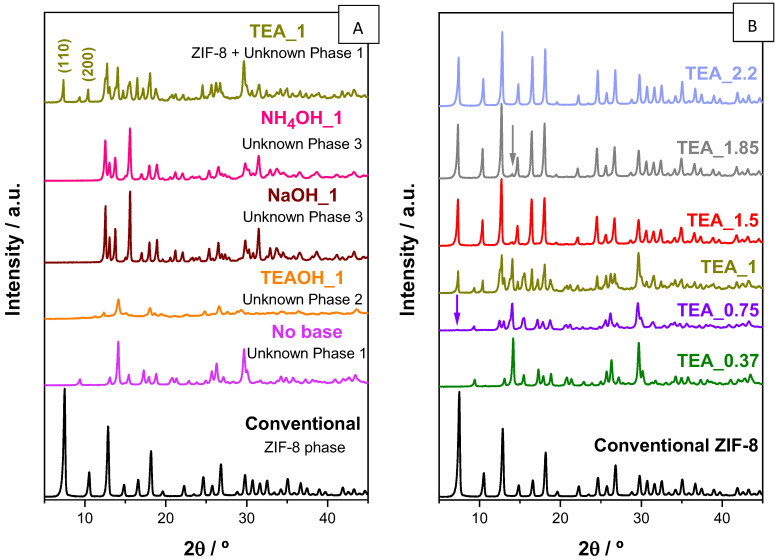
(**A**) XRD patterns of the solids resulting from the attempts of preparing ZIF-8 at room temperature from aqueous solutions with linker/Zn ratio of 20 by a conventional method [16] (black line) and with a linker/Zn ratio of 2 in the presence of the bases: TEAOH, NaOH, NH_4_OH and TEA, all of them with a base/linker ratio of 1. The detected phases are indicated. The Miller index of the lowest angle reflections of ZIF-8 are indicated on the XRD pattern of the sample TEA_1. (**B**) XRD patterns of the solids resulting from the attempts of preparing ZIF-8 at room temperature from aqueous solutions with a linker/Zn ratio of 2 in the presence of different content of TEA. The violet arrow points to the first indications of the presence of ZIF-8, whereas the gray arrow indicates the most significant peak of the Unknown phase 1, when it appears as impurity.

**Figure 2 nanomaterials-14-00348-f002:**
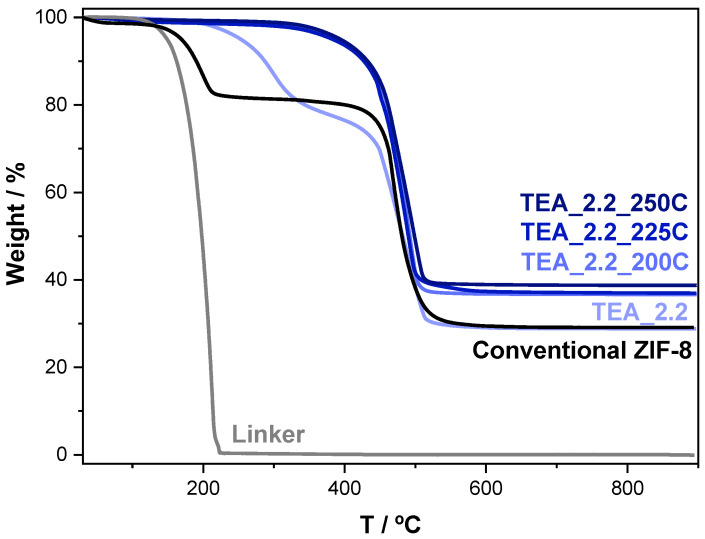
TGA profiles of the ZIF-8 samples: conventional one [16] (black line) and as-prepared one in the presence of TEA (TEA/linker ratio of 2.2) and then thermally treated at 200, 225 and 250 °C. The TGA curve of the ZIF-8 linker (2-methylimidazole) (gray curve) is shown for comparison purposes.

**Figure 3 nanomaterials-14-00348-f003:**
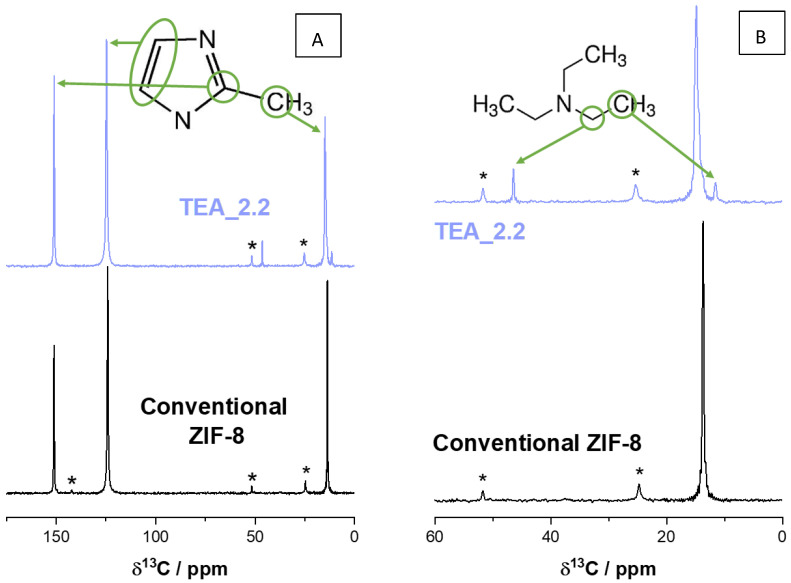
^1^H-to-^13^C CP MAS NMR spectra of the conventional ZIF-8 and the sample TEA_2.2: (**A**) full spectra and (**B**) inset of the spectra in the 0–60 ppm region. The molecules of the imidazolium and triethylamine are included for clarifying the assignment mentioned in the text. The asterisks indicate spinning side bands.

**Figure 4 nanomaterials-14-00348-f004:**
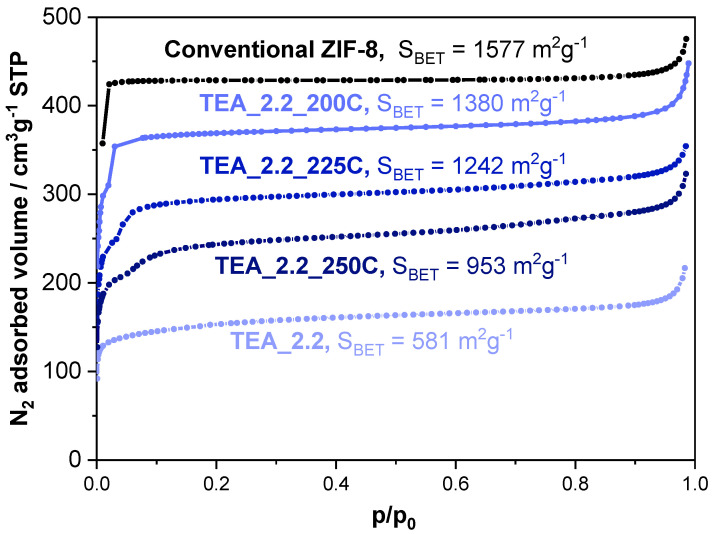
N_2_ adsorption isotherms at −196 °C of the ZIF-8 samples: conventional (black) and TEA_2 before and after different thermal treatments (different shades of blue). The BET surface area values are indicated for each isotherm.

**Figure 5 nanomaterials-14-00348-f005:**
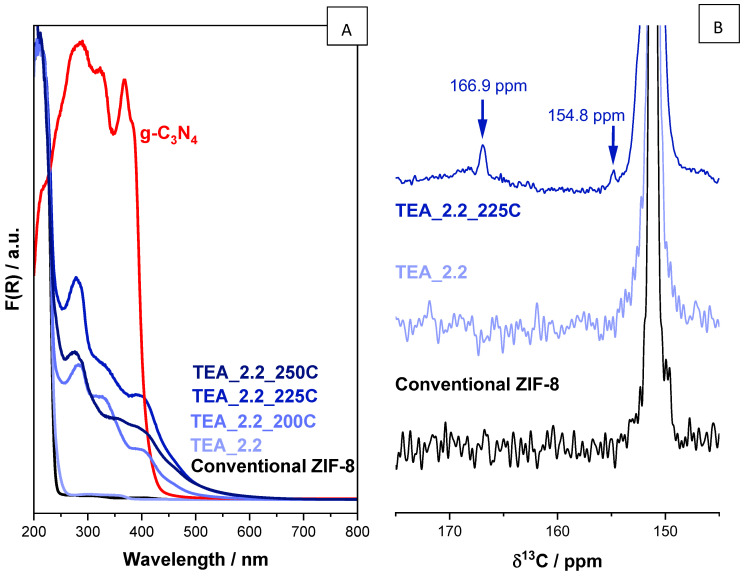
(**A**) Normalized UV-vis DRS spectra of the ZIF-8 samples: conventional (black) and TEA_2.2, TEA_2.2_200C, TEA_2.2_225C and TEA_2.2_250C (different shades of blue). The spectra of the g-C_3_N_4_ (red line) is shown for comparison purposes. (**B**) The 145–175 ppm region of the normalized ^1^H-to-^13^C CP MAS NMR spectra of the conventional ZIF-8 and the sample TEA_2.2 before and after being thermally treated at 225 °C.

**Figure 6 nanomaterials-14-00348-f006:**
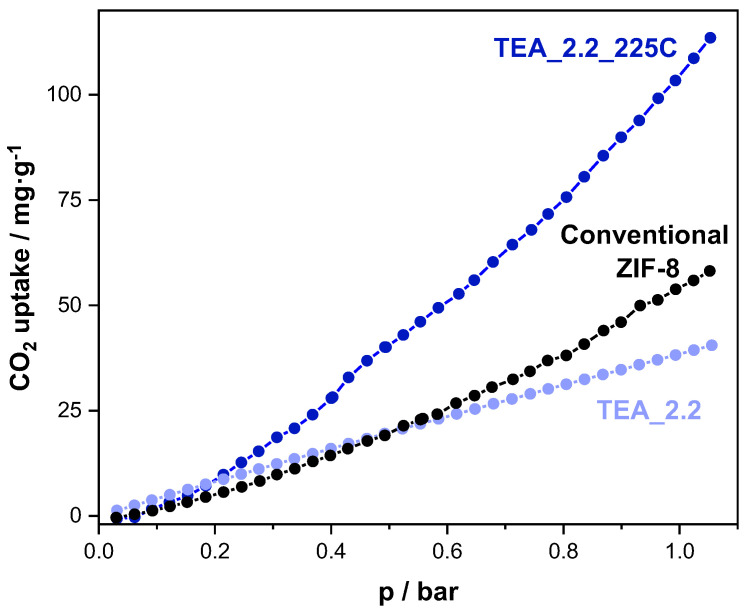
CO_2_ adsorption isotherm at 0 °C of the ZIF-8 samples: conventional (black), TEA_2.2 (light blue), TEA_2.2_225C (blue).

**Figure 7 nanomaterials-14-00348-f007:**
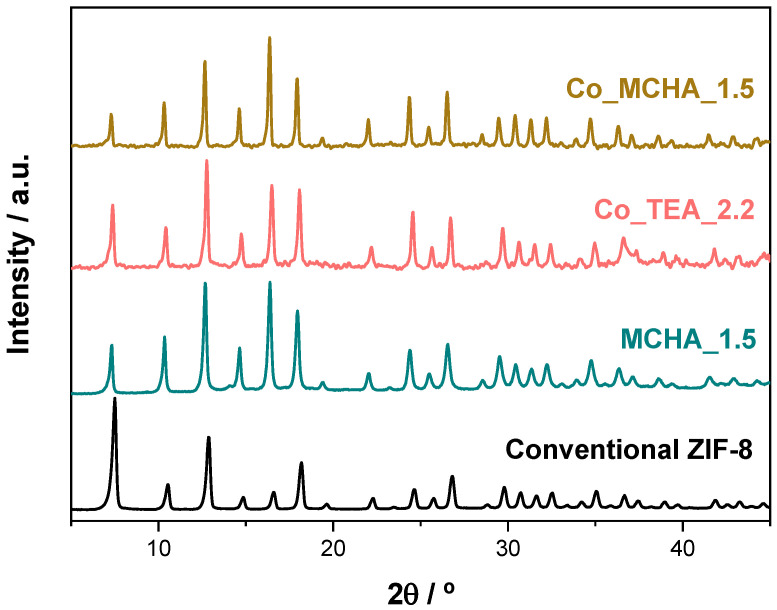
XRD patterns of the samples ZIF-8 (conventional and MCHA_1-5) and ZIF-67 (Co_TEA_2.2 and Co_MCHA_1.5).

**Figure 8 nanomaterials-14-00348-f008:**
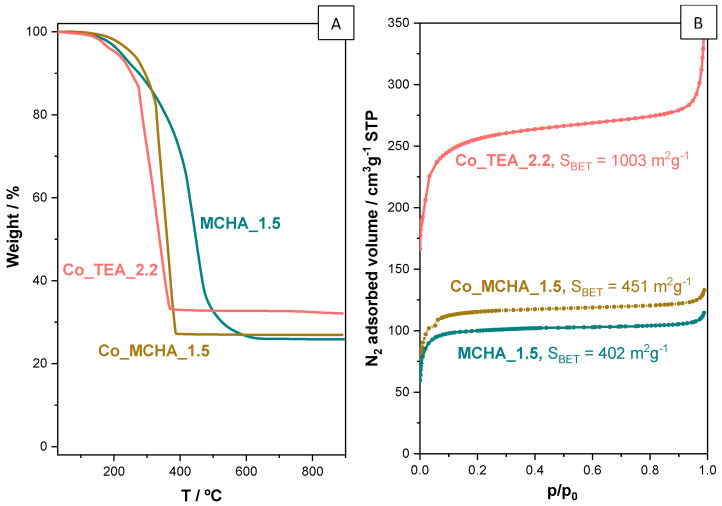
(**A**) TGA profiles and (**B**) N_2_ adsorption isotherms at −196 °C of a ZIF-8 (sample MCHA_1.5) and two ZIF-67 (samples Co_TEA_2.2 and Co_MCHA_1.5) materials prepared in the presence of amines.

## Data Availability

Data are contained within the article.

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
