# Peer review of "Sustainable Synthesis of Zeolitic Imidazolate Frameworks at Room Temperature in Water with Exact Zn/Linker Stoichiometry"

_nanomaterials, 2024, doi:10.3390/nano14040348_

Round 1

Reviewer 1 Report

Comments and Suggestions for Authors

This article delves into the development of sustainable syntheses of ZIFs. Strategies are also presented to obtain ZIF compounds with g-C3N4 that may have application. The following minor details should be corrected/commented on in this manuscript:

 -          Comment in the introduction strategies for the sustainable synthesis of ZIF-8 in the literature.

-          - Explain how the synthesis yield is calculated, if the TGA for TEA is taken into account

-          Compare CO2 adsorption values from the literature for ZIF-8 with those obtained here.

-          Take into account other TEA studies on the synthesis of ZIF-8 such e.g. [A-B]

[A] Nordin et al, Aqueous room temperature synthesis of zeolitic imidazole framework 8 (ZIF-8) with various concentrations of triethylamine, RSC Adv., 2014,4, 33292-33300

[B] Li et al, Synthesis of ZIF-8 and ZIF-67 using mixed-base and their dye adsorption, Microporous and Mesoporous Materials 234 (2016) 287-292

Author Response

First of all, we would like to thank Reviewer#1 for his/her interest in reading our manuscript, for the effort made in the evaluation and for their comments, which undoubtedly have helped us to improve the original version.

This article delves into the development of sustainable syntheses of ZIFs. Strategies are also presented to obtain ZIF compounds with g-C3N4 that may have application. The following minor details should be corrected/commented on in this manuscript:

 -          Comment in the introduction strategies for the sustainable synthesis of ZIF-8 in the literature.

Some sustainable strategies for the preparation of ZIF-8 were already included in the already submitted version of the manuscript. Six references (from 15 to 20) were mentioned and discussed in the context of the sustainable synthesis of ZIF-8. All of them describe the preparation of ZIF-8 at room temperature, most of them in water as the unique solvent, and one of them (20) even starting from a mixture with the linker/Zn stoichiometry corresponding to the linker/Zn ratio found in ZIF-8, one of the aims of our manuscript. These references are not only mentioned but also discussed in terms of their sustainability.

In any case, in our opinion, the comment by Reviewer#1 is very pertinent as our manuscript indeed focuses on the sustainable synthesis of ZIF-8. That is why we have included some other references (10 new references) related to the sustainable preparation of ZIF-8 and the references recommended by Reviewer#1 in which ZIF-8 is prepared in the presence of amines. Other included references address the formation of ZIF-8 in the presence of hydroxide ammonium, under mechanochemical procedures, etc. In particular, the following sentences have been included in page 2 of the new manuscript:

“Alternatively, ZIF-8 could be prepared under so sustainable conditions in the presence of triethylamine still with an excess of linker [21–24] or with a huge amount of ammonium hydroxide (NH4OH/linker =16) [25,26]. Other amines have been used in the synthesis of ZIF-8 starting from an excess of Zn [27]. Other quite sustainable methodologies for preparation of ZIF-8 has been also published elsewhere [28–30] but some requiring mechanical energy input.”

- Explain how the synthesis yield is calculated, if the TGA for TEA is taken into account

The yield of the synthesis of ZIF-8 has been calculated indeed from TGA. The residual weight after the TGA heating treatment under air flow is ZnO, as it was checked by XRD characterization. The corresponding Zn amount is divided by the added Zn into the synthesis mixture and multiplied by 100 to get the % yield.

That information was included in the next version of the manuscript, at the end of section 2.1.

-          Compare CO2 adsorption values from the literature for ZIF-8 with those obtained here.

Following the Reviewer#1´s advice, we have compared the CO2 uptakes of our ZIF-8 with these given in the literature for other ZIF-8. The new plot of Figure 6, where the CO2 uptake is represented in mg·g-1 and the pressure is represented as absolute pressure in bars (also following an advice by Reviewer#2) makes easier this comparison. Special care has been taken to select publications in which: (i) CO2 isotherms were registered at 0 ºC (273 K), (ii) the CO2 uptakes values is given at 1.0 bar, and (iii) the BET surface area is also given. The conclusion is clear for the non-modified ZIF-8: CO2 uptakes clearly depends on the surface area of ZIF-8 materials, as expected. Our conventional ZIF-8 uptakes the expected amount of CO2 under these conditions. Of course, the samples after removing TEA and after generating some g-C3N4 species (sample TEA_2.2_225C) has much more capability of adsorbing CO2 than any other published ZIF-8, even if they are much more porous.

All these ideas (and four more citations) have been included in the new version of the manuscript.  

-          Take into account other TEA studies on the synthesis of ZIF-8 such e.g. [A-B]

[A] Nordin et al, Aqueous room temperature synthesis of zeolitic imidazole framework 8 (ZIF-8) with various concentrations of triethylamine, RSC Adv., 2014,4, 33292-33300

[B] Li et al, Synthesis of ZIF-8 and ZIF-67 using mixed-base and their dye adsorption, Microporous and Mesoporous Materials 234 (2016) 287-292

As mentioned in the reply to the first comment by Reviewer#1, these references have been already included and discussed.

Reviewer 2 Report

Comments and Suggestions for Authors

The manuscript by Sanchez-Sanchez and co-workers entitled “PSustainable synthesis of ZIFs at room temperature in water with exact Zn/linker stoichiometry” describes the synthetic studies on iconic metal-organic framework - ZIF-8. The authos introduce various bases during the synthesis of ZIF-8 and ZIF-67 (Co-based analogue).
The application of amine as a base results in entrapping amine molecules inside the MOF pores.
The novelty of this work is rather limited, the synthesis of ZIFs in the presence of amine has been shown previously. This work, however, presents some interesting results (thermal transformation of entrapped amines into g-C3N4-like species) and could be published after revision.
Below are some comments on the manuscript structure:
- tha artcile is too long, the introductory section is chaotic and overloaded by the discussion of pH influence on MOF synthesis
- the name "N,N-methyldicyclohexylamine" is incorrect, the correct form should be "N,N-dicyclohexylmethylamine"
- the CO2 isotherms should be presented in absolute pressure values rather than in p/p0
- the discussion section should be organized better, e.g. some sentences: "The reason behind this a priori anomalous behavior is not clear at the moment (the experiment was repeat twice with the same trends)" do no add any significant value to it, however make the text very difficult for the reader. The used language should be more scientific.
In summary, this manuscript presents some interesting results and could be published in Nanomaterials  after major revision.

Comments on the Quality of English Language

please use scientific language

Author Response

Thanks to Reviewer#2 for his/her interest in reading our manuscript, for the effort made in the evaluation and for their comments, which undoubtedly have helped us to improve the original version.

The manuscript by Sanchez-Sanchez and co-workers entitled “PSustainable synthesis of ZIFs at room temperature in water with exact Zn/linker stoichiometry” describes the synthetic studies on iconic metal-organic framework - ZIF-8. The authos introduce various bases during the synthesis of ZIF-8 and ZIF-67 (Co-based analogue).
The application of amine as a base results in entrapping amine molecules inside the MOF pores.
The novelty of this work is rather limited, the synthesis of ZIFs in the presence of amine has been shown previously. This work, however, presents some interesting results (thermal transformation of entrapped amines into g-C3N4-like species) and could be published after revision.

We understand the comment by Reviewer#2 when he/she said that the novelty of the manuscript in terms of synthesis of ZIF-8 in the presence of the amines. He/she is indeed right. In fact, in the revision version, we have included some new references and discussion about the previous works in this sense. Nevertheless, our work is the only one in which ZIF-8 is prepared in water at room temperature, starting from mixtures with the linker/metal ratio corresponding to that of ZIF-8 (and ZIF-67), with an acceptable amount of amine, and with yields near 100 %. Moreover, it has been demonstrated that other bases like NaOH or ammonium hydroxide added in the same ratios cannot lead to the formation of these MOFs. For instance, in the case of ammonium hydroxide, as reported elsewhere, the required NH4OH/linker ratio is as high as 16.

Furthermore, some works focused on this kind of synthesis of ZIFs in the presence of amines have not even mentioned the main drawback of this methodology: the presence of amine within the pores, of course making these ZIFs to have lower textural properties than these of the conventional ZIFs. In addition, our study shows how ZIF-8 material could become as porous as the conventional ZIF-8 by a moderate and simple thermal treatment, which in parallel provokes the formation of nitride carbide species of high interest for certain applications, as Reviewer#2 mentioned. Even one of these possible applications of this amine-free ZIFs is made clear in our work, as the CO2 adsorption is improved against the conventional ZIF-8.

In summary, we think that our manuscript possesses enough novelty to be published in the journal Nanomaterials.  

Below are some comments on the manuscript structure:
- tha artcile is too long, the introductory section is chaotic and overloaded by the discussion of pH influence on MOF synthesis

In our opinion, the influence of pH should be addressed in the introduction. The presence of amine drastically changes the linker/Zn ratio necessary to prepare pure ZIF-8 in water at room temperature, from a ratio of around 40 to the desired ratio of 2. Since amines are bases (and because of a similar effect is found with the less basic ammonium hydroxide, therefore requiring much higher concentration), one could think that pH is a key parameter. And this is the main reason why in our work we have systematically study bases of different strength: NaOH, amines, ammonium hydroxide.

On the other hand, one of the most successful strategies to prepare carboxylate-based MOFs under sustainable conditions (room temperature, water, high yield, etc.) is precisely based on the use of the deprotonating agents, not only NaOH, but also amines or ammonium hydroxide. A discussion on applying similar strategies to these two families of different MOFs is justified, in our opinion.

In any case, we appreciate the opinion by Reviewer#2, and accordingly we have modified the introduction section following his/her advice. In particular, three long sentences (ten lines) were removed from this discussion about pH. Instead, Introduction was reinforced in the subject of previous works focused on the use of amines in the sustainable synthesis of ZIF-8, also following an advice given by Reviewer#1.

- the name "N,N-methyldicyclohexylamine" is incorrect, the correct form should be "N,N-dicyclohexylmethylamine".

Reviewer#2 is right. Thank you for this comment. We have corrected the nomenclature in the manuscript.

- the CO2 isotherms should be presented in absolute pressure values rather than in p/p0

Reviewer#2 is right. It is more convenient to plot the CO2 isotherms in absolute pressure. Thank you for this comment. We have corrected in the manuscript. In addition, we have also changed the units of Y-axis to make easier the comparison with other CO2 isotherms of ZIF-8, as asked by Reviewer#1.

- the discussion section should be organized better, e.g. some sentences: "The reason behind this a priori anomalous behavior is not clear at the moment (the experiment was repeat twice with the same trends)" do no add any significant value to it, however make the text very difficult for the reader. The used language should be more scientific.

We do not completely agree. On the one hand, the presence or the absence of this kind of sentences is not related to the organization of the manuscript.

On the other hand, in our opinion, a manuscript should not only contain results. These results should be discussed at some extent. And if an anomalous or an unexpected result is found and there is no explanation for that, readers will probably appreciate realizing that the authors encountered the same difficulties as they did in interpreting certain results or trends, rather than the authors ignoring or omitting any comments on obvious anomalies.

In any case, we perfectly understand the Reviewer#2´s point of view (surely shared with some other scientists) and accordingly we have modified some sentences in the manuscript, including that selected as an example by Reviewer#2.

In summary, this manuscript presents some interesting results and could be published in Nanomaterials after major revision.

We particularly appreciate the comments by Reviewer#2, because they were constructively critical and inspiring.  

Reviewer 3 Report

Comments and Suggestions for Authors

In this paper, ZIF-8 was synthesized using various bases, which has the advantage of reducing the amount of ligand compared to conventional methods. In addition, the g-C3N4-like formed by removing TEA from the synthesized ZIF-8 showed the effect of increasing the amount of CO2 adsorption. This is an interesting study, and we would like to recommend it for publication after carefully considering the following suggestions and comments:

1. the authors state that the amines used in the synthesis do not seem to be used as structural-directing agents, and as evidence for this, it would be helpful to provide pore size distribution data analyzed at the N2 isotherm for the various amines used.

 2. line 23, 169, 377, 486, 476 CO2, NH4OH subscripts

3. line 465 1003 m2g-1 unit subscript

Author Response

We would like to thank Reviewer#3 for his/her interest in reading our manuscript, for the effort made in the evaluation and for their comments, which undoubtedly have helped us to improve the original version of the manuscript and to better understand this investigation.

In this paper, ZIF-8 was synthesized using various bases, which has the advantage of reducing the amount of ligand compared to conventional methods. In addition, the g-C3N4-like formed by removing TEA from the synthesized ZIF-8 showed the effect of increasing the amount of CO2 adsorption. This is an interesting study, and we would like to recommend it for publication after carefully considering the following suggestions and comments:

  1. the authors state that the amines used in the synthesis do not seem to be used as structural-directing agents, and as evidence for this, it would be helpful to provide pore size distribution data analyzed at the N2 isotherm for the various amines used.

When these isotherms were registered, we did no have any intention of analyzing the pore size distribution. Because of that, the isotherms have not enough points at low partial pressure to get a well-described and trustable pore size distribution.

In any case, we do not really understand how the pore size distributions (PSD) can help us to understand the structure-directing agent (SDA) role of amine to lead to the crystallization of the ZIF-8/ZIF-67 materials. We think that such role can be discarded considering that the nature of the phase did not change when we use TEA o MCHA amines, which are very different in terms of volume and shape. In the same sense, some new references added in the introduction used different amines in the synthesis in ZIF-8. In spite of their variety, all of them led to the formation of ZIF-8, which by the way can be also formed in the absence of any amine, by just starting from a very high linker/ratio, certifying that amines has no a SDA role and suggesting that its role is more related to their base character which would induce the deprotonation of the imidazolate-based linker. 

At most, PSD curves can give us an indirect suggestion that amine is inside of the pores. But we do not have any doubt that amine molecules are indeed there. TGA-MS investigation is quite clear in this sense: TEA in inside of the ZIF-8-TEA samples and that is the reason why the surface area of our ZIF-8 materials is lower compared with that of the conventional ZIF-8. Indeed, when our ZIF-8 are heated at 200 ºC or above, practically all textural properties of ZIF-8 are recovered, indicating that TEA has left the ZIF-8 pores.

  1. line 23, 169, 377, 486, 476 CO2, NH4OH subscripts

Thanks. They have been corrected

  1. line 465 1003 m2g-1 unit subscript

Thanks. It has been corrected

Round 2

Reviewer 1 Report

Comments and Suggestions for Authors

The authors have responded correctly to the reviewers' questions.

Reviewer 2 Report

Comments and Suggestions for Authors

The authors responded to reviewer's comments and included corrections in the revised version of the manuscript. This referee is satisfied with the authors' responses and corrections.

Reviewer 3 Report

Comments and Suggestions for Authors

The request was well explained in the literature and the relevant literature was attached. We recommend publication in its current state without additional data.